# Cytotoxic Metabolites from *Calophyllum tacamahaca* Willd.: Isolation and Detection through Feature-Based Molecular Networking

**DOI:** 10.3390/metabo13050582

**Published:** 2023-04-23

**Authors:** Elise Gerometta, Gaëtan Herbette, Elnur Garayev, Arnaud Marvilliers, Jean-Valère Naubron, Carole Di Giorgio, Pierre-Eric Campos, Patricia Clerc, Allison Ledoux, Michel Frederich, Béatrice Baghdikian, Isabelle Grondin, Anne Gauvin-Bialecki

**Affiliations:** 1Laboratoire de Chimie et de Biotechnologie des Produits Naturels, Faculté des Sciences et Technologies, Université de La Réunion, Campus du Moufia, 97744 St Denis, France; elise.gerometta@univ-reunion.fr (E.G.); arnaud.marvilliers@univ-reunion.fr (A.M.); pierre-eric.campos@univ-orleans.fr (P.-E.C.); patricia.clerc@univ-reunion.fr (P.C.); isabelle.grondin@univ-reunion.fr (I.G.); 2Spectropole, FSCM, Centrale Marseille, CNRS, Aix-Marseille Université, Campus de St Jérôme–Service 511, 13397 Marseille, France; gaetan.herbette@univ-amu.fr (G.H.); jean-valere.naubron@univ-amu.fr (J.-V.N.); 3IMBE, CNRS, IRD, Aix Marseille Université, Faculté de Pharmacie, Service de Pharmacognosie, 13331 Marseille, France; elnur.garayev@univ-amu.fr (E.G.); beatrice.baghdikian@univ-amu.fr (B.B.); 4IMBE, CNRS, IRD, Aix-Marseille Université, Faculté de Pharmacie, Service de Mutagénèse Environnementale, 13385 Marseille, France; carole.di-giorgio@univ-amu.fr; 5Institut de Chimie Organique et Analytique, UMR 6759, Université d’Orléans–CNRS, Pôle de Chimie, Rue de Chartres, BP6759, CEDEX 2, 45067 Orléans, France; 6Laboratoire de Pharmacognosie, Centre Interfacultaire de Recherche sur le Médicament (CIRM), Université de Liège, Département de Pharmacie, Campus du Sart-Tilman, Quartier Hôpital, B-4000 Liège, Belgium; allison.ledoux@uliege.be (A.L.); m.frederich@uliege.be (M.F.)

**Keywords:** *Calophyllum tacamahaca*, xanthones, triterpenes, cytotoxicity, feature-based molecular networking

## Abstract

Isocaloteysmannic acid (**1**), a new chromanone, was isolated from the leaf extract of the medicinal species *Calophyllum tacamahaca* Willd. along with 13 known metabolites belonging to the families of biflavonoids (**2**), xanthones (**3**–**5**, **10**), coumarins (**6**–**8**) and triterpenes (**9**, **11**–**14**). The structure of the new compound was characterized based on nuclear magnetic resonance (NMR), high-resolution electrospray mass spectrometry (HRESIMS), ultraviolet (UV) and infrared (IR) data. Its absolute configuration was assigned through electronic circular dichroism (ECD) measurements. Compound (**1**) showed a moderate cytotoxicity against HepG2 and HT29 cell lines, with IC_50_ values of 19.65 and 25.68 µg/mL, respectively, according to the Red Dye method. Compounds **7**, **8** and **10**–**13** exhibited a potent cytotoxic activity, with IC_50_ values ranging from 2.44 to 15.38 µg/mL, against one or both cell lines. A feature-based molecular networking (FBMN) approach led to the detection of a large amount of xanthones in the leaves extract, and particularly analogues of the cytotoxic isolated xanthone pyranojacareubin (**10**).

## 1. Introduction

The genus *Calophyllum* (Calophyllaceae) includes approximately 200 species, distributed across all tropical regions. They are traditionally used against many ailments, including ulcers, malaria, tumor, infections, eye diseases, pain, inflammation and rheumatism [1,2]. This genus is an important source of bioactive natural products, including coumarins, xanthones, chromanones and triterpenes [3,4]. Xanthones and coumarins from *Calophyllum* species are known to possess cytotoxic, antiviral, antimicrobial, antiparasite, analgesic, anti-inflammatory and chemopreventive properties [5,6]. (+)-Calanolide A, a pyranocoumarin isolated from *C. lanigerum*, reached phase II of a clinical trial for its potent inhibitory activity of HIV-1 reverse transcriptase [4].

The species *Calophyllum tacamahaca* Willd., commonly known as “Takamaka des Hauts”, is an endemic tree to Mauritius and Reunion Island. The leaf species is registered in the List of plants used in traditional medicine of French pharmacopoeia since April 2022, for eye diseases, fever, headaches and as veinotonic. This species is also traditionally employed to treat skin diseases, memory disorders, rheumatism and blood circulation troubles [7]. Previous investigations showed that leaf extract possesses hypotensive [8], antiplasmodial [9], antimicrobial [7], antiviral [10] and anti-inflammatory [11] activities. Nevertheless, the chemical composition of the species has never been studied and so bioactive compounds of the species have never been isolated nor identified so far.

Thus, the ethyl acetate (EtOAc) leaf extract of *C. tacamahaca* was subjected to a bio-guided chemical investigation in order to identify bioactive metabolites. Herein, we report the isolation, structure characterization and in vitro cytotoxic activity of one new chromanone (**1**), along with 13 known compounds (**2**–**14**). A feature-based molecular networking (FBMN) approach was performed in order to detect analogues of the bioactive compounds, and the obtained results are discussed below.

## 2. Materials and Methods

### 2.1. General Experimental Procedures

Optical rotations were determined using an Anton Paar MCP200 polarimeter (589 nm, 25 °C), and UV spectra were acquired on a Thermo Scientific DAD spectrophotometer. IR spectra were recorded on a Vertex 70 (Bruker) ATR-FTIR spectrometer. For compound (**1**), UV–vis and experimental ECD spectra were recorded on a JASCO J-815 spectrometer equipped with a JASCO Peltier cell holder PTC-423 to maintain the temperature at 20.0 °C. The handedness of the circular polarized light was modulated at 50 kHz with a quartz photoelastic modulator set at l/4 retardation. A quartz cell of 1 mm of optical path length was used. Sample was prepared in dry methanol at a concentration of 0.0005 mol. L^−1^. ECD spectra were recorded using CD_3_OD as a reference and are presented without smoothing and further data processing. NMR spectra were acquired in CD_3_OD (δ_1H_ 3.31 ppm, δ_13C_ 49.00 ppm) on a Bruker Avance II^+^ 600 MHz (TCI cryoprobe) spectrometer at 300 K. NMR spectra were analyzed with the TopSpin (v 4.1.1) software. Structural assignments were based on ^1^H NMR, ^13^C NMR, COSY, HSQC and HMBC spectra. The chemical shifts δ are provided in ppm and coupling constants *J* in Hz. UHPLC-HRESIMS and UHPLC-HRESIMS/MS analyses were performed on an Impact II Bruker Daltonics Qq-TOF spectrometer with an ESI source using a 2.1 × 150 mm 1.6 µm RP-C_18_ column (Luna Omega C18, Phenomenex, Torrance, CA, USA) and an elution gradient of H_2_O-CH_3_CN with 0.1% HCO_2_H (98:2 to 0:100). Solid reverse-phase extraction was performed over 10 g SPE Strata 55 µm C18 tubes (Phenomenex), with three elution steps (H_2_O/CH_3_CN *v*/*v*). Preparative HPLCs were performed on a Waters 2545 system using MassLynx software and a 21.2 × 150 mm 5 µm RP-C_18_ column (Gemini C18, Phenomenex), with an appropriate elution gradient of H_2_O-CH_3_CN with 0.1% HCO_2_H at a flow rate of 20 mL/min. Semi-preparative HPLC was performed on a Dionex Ultimate 3000 system (Thermo Scientific, Waltham, MA, USA) using Chromeleon software and a 10 × 250 mm 5 µm RP-C_18_ column (Gemini C18, Phenomenex) with an appropriate elution gradient of H_2_O-CH_3_CN with 0.1% HCO_2_H at a flow rate of 4.5 mL/min. Analytical HPLC was performed on a Dionex Ultimate 3000 system (Thermo Scientific) using Chromeleon software and a 4.6 × 150 mm 3 µm RP-C_18_ column (Gemini C18, Phenomenex) with an appropriate elution gradient of H_2_O-CH_3_CN with 0.1% HCO_2_H at a flow rate of 0.8 mL/min.

### 2.2. Plant Material

Leaves of *Calophyllum tacamahaca* were collected in March 2019 on Reunion Island (Saint Denis). The taxonomic identification of the plant species was performed by Mr. H. Thomas (Parc National de La Réunion). A voucher specimen was deposited in the Herbarium of the University of La Réunion for confirmation of identification, with the following accession number: REU024075.

### 2.3. Extraction and Isolation

Leaves of *C. tacamahaca* were dried at 40 °C for 48 h and powdered. An accelerated solvent extractor (ASE 300 Dionex) was used to exhaustively extract the ground material (237.0 g). Four successive extractions were performed at 40 °C with EtOAc. The extract was evaporated under reduced pressure at 38 °C to obtain 20.3 g of crude extract.

A total of 3.32 g of crude extract were fractionated by solid reverse-phase extraction using combinations of H_2_O/CH_3_CN (*v*/*v*) of decreasing polarity. Three fractions (F1–F3) were obtained and evaluated for their cytotoxic activity against cancer cell lines. Fraction F2 (240.1 mg) was then subjected to preparative HPLC using an elution gradient of H_2_O-CH_3_CN with 0.1% HCO_2_H (55:45 over 5 min, 55:45 to 20:80 over 35 min) at a flow rate of 20 mL/min (UV 260 nm). The purification of fraction F2 afforded the pure compounds isocaloteysmannic acid (**1**, 13.1 mg), amentoflavone (**2**, 12.0 mg), 6-(4-hydroxy-3-methylbutyl)-1,5-dihydroxyxanthone (**3**, 1.7 mg), scriblitifolic acid (**4**, 1.7 mg), pancixanthone B (**5**, 1.9 mg) and isocalophyllic acid (**7**, 29.4 mg). Subfraction F2–7 (11.4 mg) was subjected to semi-preparative HPLC using an isocratic elution of H_2_O-CH_3_CN with 0.1% HCO_2_H (35:65) for 18 min at a flow rate of 4.5 mL/min (UV 320 nm) and afforded the pure compounds isocalophyllic acid (**7**, 1.0 mg) and inophyllum E (**8**, 2.8 mg). The purification of subfraction F2–8 (13.7 mg) was performed by semi-preparative HPLC using an isocratic elution of H_2_O-CH_3_CN with 0.1% HCO_2_H (35:65) for 20 min at a flow rate of 4.5 mL/min (UV 280 nm) and yielded the pure compound calophyllic acid (**6**, 2.5 mg). Fraction F3 (673.3 mg) was subjected to preparative HPLC using an elution gradient of H_2_O-CH_3_CN with 0.1% HCO_2_H (30:70 over 2 min, 30:70 to 0:100 over 10 min, 0:100 over 9 min) at a flow rate of 20 mL/min (ELSD). The purification of fraction F3 afforded the pure compounds isocalophyllic acid (**7**, 35.8 mg), calophyllic acid (**6**, 15.9 mg), canophyllalic acid (**11**, 22.8 mg), canophyllol (**12**, 21.9 mg) and canophyllic acid (**13**, 25.2 mg). The purification of subfraction F3–6 (5.7 mg) was performed by analytical HPLC using an isocratic elution of H_2_O-CH_3_CN with 0.1% HCO_2_H (30:70) for 16 min at a flow rate of 0.8 mL/min (CAD) and yielded the pure compound 27-hydroxyacetate-canophyllic acid (**9**, 0.5 mg). Subfraction F3–7 (5.6 mg) was subjected to analytical HPLC using an isocratic elution of H_2_O-CH_3_CN with 0.1% HCO_2_H (22:78) for 13 min at a flow rate of 0.8 mL/min (UV 280 nm) and yielded the pure compound pyranojacareubin (**10**, 0.9 mg). The purification of subfraction F3–19 was performed by semi-preparative HPLC using an isocratic elution of H_2_O-CH_3_CN with 0.1% HCO_2_H (10:90) for 32 min at a flow rate of 4.5 mL/min (ELSD) and yielded the pure compound canophyllal (**14**, 0.3 mg).

**Isocaloteysmannic acid** (**1**): yellow–green powder, ${\left[\alpha \right]}_{D}^{25}$ −31.7 (*c* 0.1, MeOH); UV (MeOH) λ_max_ 200, 264–274, 299–312, 368 nm; IR ν_max_ 3087, 2977, 2926, 2855, 1709, 1627, 1300, 1000 cm^−1^; for ^1^H and ^13^C NMR spectroscopic data, see Table 1; HRESIMS *m*/*z* 423.1791 [M+H]^+^ (calculated for C_25_H_27_O_6_^+^, 423.1802).

**Amentoflavone** (**2**): yellow powder, UV (MeOH) λ_max_ 220, 272, 332 nm; ^1^H NMR (CD_3_OD, 600 MHz) 7.96 (1H, brs), 7.84 (1H, brd, *J* = 7.8 Hz), 7.50 (2H, d, *J* = 7.8 Hz), 7.08 (1H, brd, *J* = 7.8), 6.69 (2H, d, *J* = 7.8), 6.56 (1H, s), 6.55 (1H, s), 6.37 (1H, brs), 6.34 (1H, s), 6.16 (1H, brs); ^13^C NMR (CD_3_OD, 150 MHz) 184.2 (C=O), 183.8 (C=O), 166.2 (CO), 166.0 (COH, CO), 164.3 (COH), 163.2 (COH), 162.5 (COH), 161.2 (COH), 159.4 (CO), 156.5 (CO), 132.8 (CH), 129.4 (CH), 128.8 (CH), 123.3 (C), 123.2 (C), 121.9 (C), 117.7 (CH), 116.8 (CH), 105.6 (C), 105.3 (C), 104.0 (CH), 103.4 (CH), 100.4 (CH), 100.2 (CH), 95.2 (CH); HRESIMS *m*/*z* 539.0941 [M+H]^+^ (calculated for C_30_H_19_O_10_^+^, 539.0973).

**6-(4-Hydroxy-3-methylbutyl)-1,5-dihydroxyxanthone** (**3**): yellow powder, UV (MeOH) λ_max_ 250, 316, 370 nm; ^1^H NMR (CD_3_OD, 600 MHz) 7.65 (1H, brt, *J* = 8.2 Hz), 7.65 (1H, d, *J* = 8.1 Hz), 7.20 (1H, d, *J* = 8.1 Hz), 7.08 (1H, d, *J* = 8.2 Hz), 6.76 (1H, d, *J* = 8.2 Hz), 3.48 (1H, dd, *J* = 10.7, 5.9 Hz), 3.41 (1H, dd, *J* = 10.7, 6.5 Hz), 2.89 (1H, ddd, *J* = 13.3, 10.0, 5.5 Hz), 2.80 (1H, ddd, *J* = 13.3, 9.0, 6.2 Hz), 1.81 (1H, m), 1.66 (1H, m), 1.46 (1H, m), 1.01 (1H, d, *J* = 6.8 Hz); ^13^C NMR (CD_3_OD, 150 MHz) 183.8 (C=O), 163.1 (COH), 157.5 (CO), 146.8 (CO), 144.6 (COH), 138.6 (C), 137.9 (CH), 126.5 (CH), 120.3 (C), 116.2 (CH), 111.2 (CH), 109.6 (C), 108.2 (CH), 68.3 (CH_2_OH), 36.8 (CH), 34.4 (CH_2_), 29.0 (CH_2_), 17.0 (CH_3_); HRESIMS *m*/*z* 315.1221 [M+H]^+^ (calculated for C_18_H_19_O_5_^+^, 315.1227).

**Scriblitifolic acid** (**4**): yellow–beige powder, UV (MeOH) λ_max_ 237, 249, 298, 366 nm; ^1^H NMR (CD_3_OD, 600 MHz) 7.84 (1H, d, *J* = 8.0 Hz), 7.63 (1H, t, *J* = 8.2 Hz), 7.25 (1H, d, *J* = 8.0 Hz), 7.02 (1H, d, *J* = 8.2 Hz), 6.75 (1H, d, *J* = 8.2 Hz), 4.03 (3H, s), 2.81 (2H, t, *J* = 7.1 Hz), 2.45 (1H, m), 1.98 (2H, m), 1.72 (1H, m), 1.21 (1H, d, *J* = 6.5 Hz); ^13^C NMR (CD_3_OD, 150 MHz) 183.4 (C=O), 182.4 (COOH), 162.9 (COH), 157.3 (CO), 151.1 (CO), 147.1 (CO), 144.8 (C), 138.1 (CH), 126.5 (CH), 121.3 (C), 121.0 (CH), 111.4 (CH), 109.6 (C), 108.3 (CH), 62.2 (OCH_3_), 42.0 (CH), 36.0 (CH_2_), 29.3 (CH_2_), 18.1 (CH_3_); HRESIMS *m*/*z* 343.1166 [M+H]^+^ (calculated for C_19_H_19_O_6_^+^, 343.1176).

**Pancixanthone B** (**5**): beige powder, UV (MeOH) λ_max_ 219, 248, 325, 363 nm; ^1^H NMR (CD_3_OD, 600 MHz) 7.61 (1H, d, *J* = 7.8 Hz), 7.24 (1H, brs), 7.18 (1H, t, *J* = 7.8 Hz), 6.15 (1H, s), 4.55 (1H, q, *J* = 6.6 Hz), 1.61 (3H, s), 1.41 (3H, d, *J* = 6.6 Hz), 1.33 (3H, s); ^13^C NMR (CD_3_OD, 150 MHz) 182.2 (C=O), 167.8 (CO), 165.3 (COH), 154.1 (CO), 148.0 (COH), 146.6 (CO), 124.9 (CH), 122.7 (C), 121.2 (CH), 116.2 (CH), 114.5 (C), 104.6 (C), 94.3 (CH), 92.5 (CH), 45.0 (C), 25.9 (CH_3_), 21.4 (CH_3_), 14.6 (CH_3_); HRESIMS *m*/*z* 313.1075 [M+H]^+^ (calculated for C_18_H_17_O_5_^+^, 313.1071).

**Calophyllic acid** (**6**): dark green powder, UV (MeOH) λ_max_ 200, 270, 320, 366 nm; ^1^H NMR (CDCl_3_, 600 MHz) 12.55 (1H, s), 7.38 (2H, m), 7.32 (1H, m), 7.30 (2H, m), 6.53 (1H, d, *J* = 9.5 Hz), 6.44 (1H, s), 5.42 (1H, d, *J* = 9.5 Hz), 4.27 (1H, dq, *J* = 11.5, 5.9 Hz), 2.63 (1H, dq, *J* = 11.5, 6.9 Hz), 1.54 (3H, d, *J* = 5.9 Hz), 1.26 (3H, s), 1.22 (3H, d, *J* = 6.9 Hz), 1.06 (3H, s); ^13^C NMR (CDCl_3_, 150 MHz) 198.7 (C=O), 170.2 (COOH), 160.5 (COH), 158.7 (CO), 156.7 (CO), 149.7 (C), 140.8 (C), 129.3 (CH), 128.5 (CH), 127.3 (CH), 126.3 (CH), 120.1 (CH), 115.6 (CH), 108.0 (C), 101.7 (C), 101.4 (C), 79.1 (CH), 78.4 (C), 45.8 (CH), 28.4 (CH_3_), 28.2 (CH_3_), 19.9 (CH_3_), 10.1 (CH_3_); HRESIMS *m*/*z* 421.1651 [M+H]^+^ (calculated for C_25_H_25_O_6_^+^, 421.1646).

**Isocalophyllic acid** (**7**): dark green powder, UV (MeOH) λ_max_ 200, 270, 320, 366 nm; ^1^H NMR (CD_3_OD, 600 MHz) 7.35 (2H, m), 7.31 (3H, m), 6.56 (1H, d, *J* = 10.1 Hz), 6.43 (1H, s), 5.49 (1H, d, *J* = 10.1 Hz), 4.68 (1H, qd, *J* = 6.9, 3.8 Hz), 2.65 (1H, qd, *J* = 7.3, 3.8 Hz), 1.44 (3H, d, *J* = 6.9 Hz), 1.29 (3H, s), 1.19 (3H, d, *J* = 7.3 Hz), 0.97 (3H, s); ^13^C NMR (CD_3_OD, 150 MHz) 202.8 (C=O), 169.9 (COOH), 162.0 (COH), 160.0 (CO), 157.7 (CO), 148.6 (C), 142.4 (C), 129.8 (CH), 129.3 (CH), 128.1 (CH), 127.4 (CH), 122.6 (CH), 116.5 (CH), 109.8 (C), 102.7 (C), 102.1 (C), 79.4 (C), 78.0 (CH), 45.6 (CH), 28.7 (CH_3_), 28.2 (CH_3_), 16.7 (CH_3_), 9.7 (CH_3_); HRESIMS *m/z* 421.1639 [M+H]^+^ (calculated for C_25_H_25_O_6_^+^, 421.1646).

**Inophyllum E** (**8**): yellow powder, UV (MeOH) λ_max_ 200, 270, 310, 366 nm; ^1^H NMR (CD_3_OD, 600 MHz) 7.28 (2H, m), 7.23 (2H, m), 7.22 (1H, m), 6.49 (1H, d, *J* = 10.0 Hz), 6.02 (1H, s), 5.46 (1H, d, *J* = 10.0 Hz), 4.65 (1H, qd, *J* = 6.5, 3.4 Hz), 2.64 (1H, qd, *J* = 7.4, 3.4 Hz), 1.41 (3H, d, *J* = 6.5 Hz), 1.18 (3H, d, *J* = 7.4 Hz), 1.04 (3H, s), 0.99 (3H, s); ^13^C NMR (CD_3_OD, 150 MHz) 202.7 (C=O), 162.0 (OC=O), 160.5 (CO), 157.7 (CO), 141.8 (C), 129.9 (CH), 128.4 (CH), 128.3 (CH), 127.7 (CH), 125.2 (CH), 146.4 (C), 116.3 (CH), 113.3 (C), 102.9 (C), 101.9 (C), 79.3 (C), 77.9 (CH), 45.5 (CH), 27.9 (CH_3_), 27.7 (CH_3_), 16.5 (CH_3_), 9.7 (CH_3_); HRESIMS *m/z* 403.1529 [M+H]^+^ (calculated for C_25_H_23_O_5_^+^, 403.1540).

**27-Hydroxyacetate-canophyllic acid** (**9**): yellow powder, ^1^H NMR (CDCl_3_, 600 MHz) 4.43 (1H, d, *J* = 12.2 Hz), 4.34 (1H, d, *J* = 12.2Hz), 3.73 (1H, m), 2.43 (1H, m), 2.21 (1H, m), 2.07 (1H, s), 1.96 (1H, m), 1.72 (1H, m), 1.55 (1H, m), 1.51 (2H, m), 1.50 (1H, m), 1.39 (1H, m), 1.38 (1H, m), 1.36 (1H, m), 1.34 (1H, m), 1.32 (1H, m), 1.31 (1H, m), 1.30 (1H, brs), 1.26 (1H, m), 1.24 (2H, m), 1.14 (1H, m), 1.11 (1H, m), 1.02 (3H, s), 0.96 (3H, s), 0.92 (3H, d, *J* = 7.3 Hz), 0.90 (3H, s), 0.89 (3H, s), 0.85 (1H, brs), 0.85 (3H, s); ^13^C NMR (CDCl_3_, 150 MHz) 182.9 (COOH), 171.7 (OC=O), 72.7 (COH), 65.4 (CH_2_), 61.3 (CH), 53.2 (CH), 49.0 (CH), 44.6 (C), 42.2 (C), 41.3 (CH_2_), 38.4 (CH), 38.1 (C), 37.8 (C), 37.5 (C), 36.3 (CH_2_), 36.1 (CH_2_), 35.7 (CH_2_), 34.7 (CH_3_), 32.6 (CH_2_), 31.9 (CH_2_), 29.7 (CH_2_), 29.6 (CH_3_), 28.3 (C), 25.1 (CH_2_), 21.6 (CH_3_), 21.4 (CH_3_), 18.6 (CH_2_), 17.9 (CH_3_), 16.5 (CH_3_), 15.9 (CH_2_), 11.9 (CH_3_).

**Pyranojacareubin** (**10**): yellow powder, UV (MeOH) λ_max_ 200, 290–300, 350 nm; ^1^H NMR (CDCl_3_, 600 MHz) 13.30 (1H, s), 7.47 (1H, s), 6.72 (1H, d, *J* = 10.3 Hz), 6.43 (1H, s), 6.43 (1H, d, *J* = 10.5 Hz), 5.73 (1H, d, *J* = 10.5 Hz), 5.59 (1H, d, *J* = 10.3 Hz), 1.53 (6H, s), 1.47 (6H, s); ^13^C NMR (CDCl_3_, 150 MHz) 180.0 (C=O), 160.4 (CO), 157.8 (COH), 157.2 (CO), 145.1 (CO), 132.1 (COH), 131.2 (CH), 127.7 (CH), 121.5 (CH), 117.8 (C), 115.6 (CH), 113.7 (CH), 104.8 (C), 103.3 (C), 95.4 (CH), 79.1 (C), 78.2 (C), 28.6 (CH_3_), 28.5 (CH_3_).

**Canophyllalic acid** (**11**): green powder, ^1^H NMR (CDCl_3_, 600 MHz) 2.40 (1H, brdd, *J* = 13.6, 4.3 Hz), 2.39 (1H, m), 2.34 (1H, m), 2.28 (1H, m), 2.23 (1H, q, *J* = 6.9 Hz), 1.95 (1H, m), 1.74 (1H, m); 1.68 (1H, m), 1.67 (1H, m), 1.52 (1H, brdd, *J* = 12.7, 2.7 Hz), 1.51 (1H, m), 1.49 (1H, m), 1.476 (1H, m), 1.472 (1H, m), 1.44 (1H, m), 1.42 (1H, m), 1.41 (1H, brdd, *J* = 10.1, 2.2), 1.39 (1H, m), 1.35 (1H, brt, *J* = 13.6 Hz), 1.29 (1H, m), 1.27 (1H, m), 1.25 (1H, m), 1.20 (1H, m), 1.195 (1H, m), 1.17 (1H, brdd, *J* = 13.6, 4.3 Hz), 1.04 (3H, s), 1.03 (3H, s), 0.94 (3H, s), 0.87 (3H, d, *J* = 6.9 Hz), 0.86 (3H, s), 0.81 (3H, s), 0.71 (3H, s); ^13^C NMR (CDCl_3_, 150 MHz) 213.3 (C=O), 184.9 (COOH), 59.4 (CH), 58.4 (CH), 53.2 (CH), 45.0 (C), 42.2 (C), 41.6 (CH_2_), 41.3 (CH_2_), 39.1 (C), 38.0 (CH), 37.9 (C), 37.8 (C), 36.1 (CH_2_), 35.6 (CH_2_), 35.0 (CH_2_), 34.7 (CH_3_), 32.8 (CH_2_), 32.6 (CH_2_), 31.2 (CH_2_), 29.9 (CH_3_), 29.6 (CH_2_), 28.6 (C), 22.4 (CH_2_), 20.8 (CH_3_), 18.7 (CH_3_), 18.3 (CH_2_), 17.7 (CH_3_), 14.8 (CH_3_), 7.0 (CH_3_).

**Canophyllol** (**12**): green powder, ^1^H NMR (CDCl_3_, 600 MHz) 3.64 (1H, d, *J* = 11.9 Hz), 3.61 (1H, d, *J* = 11, 9 Hz), 2.38 (1H, m), 2.28 (1H, m), 2.24 (1H, q, *J* = 6.6 Hz), 1.96 (1H, m), 1.84 (1H, m), 1.75 (1H, m), 1.68 (1H, m), 1.53 (1H, brdd, *J* = 12.3, 2.2 Hz), 1.48 (1H, m), 1.47 (1H, m), 1.46 (2H, m), 1.41 (2H, m), 1.35 (1H, m), 1.32 (2H, m), 1.31 (1H, m), 1.30 (1H, m), 1.29 (2H, m), 1.27 (1H, m), 1.26 (1H, m), 1.12 (3H, s), 0.99 (3H, s), 0.97 (3H, s), 0.91 (3H, s), 0.87 (3H, d, *J* = 6.6 Hz), 0.86 (3H, s), 0.71 (3H, s); ^13^C NMR (CDCl_3_, 150 MHz) 213.3 (C=O), 68.2 (COH), 59.6 (CH), 58.4 (CH), 52.6 (CH), 42.2 (C), 41.6 (CH_2_), 41.4 (CH_2_), 39.6 (CH), 39.5 (C), 38.3 (C), 37.6 (C), 35.6 (CH_2_), 35.3 (C), 34.6 (CH_2_), 34.4 (CH_3_), 33.5 (CH_2_), 33.0 (CH_3_), 31.5 (CH_2_), 31.4 (CH_2_), 30.2 (CH_2_), 29.3 (CH_2_), 28.3 (C), 22.4 (CH_2_), 19.3 (CH_3_), 19.2 (CH_3_), 18.4 (CH_2_), 18.2 (CH_3_), 14.8 (CH_3_), 7.0 (CH_3_).

**Canophyllic acid** (**13**): orange powder, ^1^H NMR (CDCl_3_, 600 MHz) 3.73 (1H, m), 2.38 (1H, brdd, *J* = 13.3, 4.0 Hz), 1.89 (1H, m), 1.73 (1H, m), 1.66 (1H, m), 1.55 (1H, m), 1.54 (1H, m), 1.50 (1H, m), 1.45 (1H, m), 1.43 (1H, m), 1.42 (1H, m), 1.35 (1H, m), 1.34 (1H, m), 1.33 (1H, m), 1.29 (1H, m), 1.25 (1H, m), 1.245 (1H, m), 1.23 (2H, m), 1.17 (1H, m), 1.13 (1H, m), 1.03 (3H, s), 1.0 (3H, s), 0.97 (1H, m), 0.96 (3H, s), 0.934 (3H, s), 0.93 (3H, d, *J* = 7.0 Hz), 0.89 (1H, m), 0.85 (3H, s), 0.80 (3H, s); ^13^C NMR (CDCl_3_, 150 MHz) 184.0 (COOH), 72.9 (COH), 61.3 (CH), 53.3 (CH), 49.3 (CH), 44.9 (CH), 41.7 (CH_2_), 39.1 (C), 38.1 (CH), 38.0 (C), 37.9 (C), 37.5 (C), 36.1 (CH_2_), 35.6 (CH_2_), 35.3 (CH_2_), 35.0 (CH_2_), 34.7 (CH_3_), 32.8 (CH_2_), 32.7 (CH_2_), 31.4 (CH_2_), 29.9 (CH_3_), 29.7 (CH_2_), 28.6 (C), 20.7 (CH_3_), 18.7 (CH_3_), 18.0 (CH_3_), 17.6 (CH_2_), 16.5 (CH_3_), 16.0 (CH_2_), 11.8 (CH_3_).

**Canophyllal** (**14**): off-white powder, ^1^H NMR (CDCl_3_, 600 MHz) 9.47 (1H, s), 2.39 (1H, ddd, *J* = 13.4, 5.1, 2.0), 2.28 (1H, tdd, *J* = 13.4, 7.4, 0.9), 2.23 (1H, q, *J* = 7.7 Hz), 2.18 (1H, dd, *J* = 13.4, 4.4 Hz), 2.01 (1H, m), 1.99 (1H, m), 1.95 (1H, m), 1.74 (1H, dt, *J* = 12.5, 3.0 Hz), 1.52 (1H, dd, *J* = 12.3, 3.0 Hz), 1.50 (1H, m), 1.46 (1H, m), 1.43 (1H, m), 1.38 (1H, m), 1.37 (1H, m), 1.25 (2H, m), 1.07 (3H, s), 0.98 (3H, s), 0.95 (3H, s), 0.87 (3H, d, *J* = 7.7 Hz), 0.84 (3H, s), 0.71 (3H, s), 0.67 (3H, s); ^13^C NMR (CDCl_3_, 150 MHz) 213.3 (C=O), 209.4 (HC=O), 59.4 (CH), 58.3 (CH), 53.0 (CH), 48.0 (C), 42.1 (C), 41.6 (CH_2_), 41.5 (CH_2_), 38.5 (C), 38.0 (C), 37.7 (C), 36.5 (C), 35.5 (CH_2_), 34.9 (CH_2_), 34.6 (CH_3_), 32.6 (CH_2_), 32.5 (CH_2_), 30.7 (CH_2_), 29.5 (CH_3_), 28.2 (C), 27.3 (CH_2_), 22.5 (CH_2_), 19.9 (CH_3_), 18.8 (CH_3_), 18.2 (CH_2_), 17.3 (CH_3_), 14.6 (CH_3_), 6.8 (CH_3_).

### 2.4. Molecular Modeling

#### 2.4.1. Calculation of Averaged NMR Spectra

The GAUSSIAN 09 program [12] using the hybrid B3LYP exchange–correlation functional [13,14] and the 6-31+G(d,p) basis set was used to carry out all DFT calculations. Tight convergence criteria were used for geometry optimization. All stationary points were confirmed as true minima via vibrational frequency calculations. Frequencies calculated in the harmonic approximation were multiplied by 0.98. Density functional theory (DFT) was used to perform the quantum chemical calculations. The molecular geometries were optimized by the DFT/B3LYP/6-31+G(d,p) method. Gauge including atomic orbitals (GIAO) NMR chemical shifts were calculated for the obtained geometries using the polarizable continuum model, PCM, with methanol as solvent, mPW1PW91 DFT functional and 6-31+G(d,p) basis sets to be in agreement with the DP4+ probability calculation. Averaged NMR chemical shifts were calculated from the unscaled chemical shifts of individual conformers according to their contribution calculated by Boltzmann weighting and using TMS as reference standard.

#### 2.4.2. Conformational Study for UV–ECD Calculations

Conformational analysis was performed by stochastic exploration of the potential energy surface (PES) using the simulated annealing algorithm proposed by the Ampac11 software and combined with semi-empirical levels RM1 [15]. For the annealing, a geometry optimized at GD3BJ-B3LYP/6-311G(d,p) level was used as a starting structure. The GD3BJ term stands for empirical dispersion which was added with the D3 version of Grimme’s dispersion with Becke–Johnson damping (GD3BJ) [16]. During each annealing, only the dihedral angles of this initial geometry were allowed to relax, the bond lengths and the valence angles were kept constant. A set of 24 geometries (the conformations with energy lower than 3 kcal mol^−1^ compared to the lower energy conformation) were selected for each diastereomer from the structures generated by 4 simulated annealing algorithms, each performed either with an initial geometry with some dihedral angles modified or with a different annealing temperature. Then, these geometries were fully optimized (i.e., all internal coordinates released) using GD3BJ-B3LYP/6-311G(d,p) level.

#### 2.4.3. Calculation of Averaged UV and ECD Spectra

Based on the GD3BJ-B3LYP/6-311G(d,p) optimized geometries, the UV and ECD spectra were calculated using time-dependent density functional theory (TDSCF-DFT) with CAM-B3LYP functional and 6-31++G(d,p) basis set and with the SMD(CH_3_OH) solvation model. SMD indicates the implicit solvent model used which is a dielectric continuum model that simulates the average effects of the solvent [17]. Calculations were performed for vertical 1A singlet excitation for 50 states. For a comparison between theoretical results and the experimental values, the calculated UV and ECD spectra have been modeled with a gaussian function using a half-width of 0.33 eV. Due to the approximations of the theoretical model used, an almost constant offset was observed between measured and calculated wavenumbers. Using UV spectra, all frequencies were calibrated by a factor of 1.05. Gaussian 16 package [18] was used to perform all calculations. It should be noted that similar calculations were performed using the LC-whPBE functional instead of CAM-B3LYP (SMD(CH_3_OH)/LC-whPBE/6-31++G(d,p)//GD3BJ-B3LYP/6-311G(d,p)) and led to a similar result, which is not presented here.

### 2.5. In Vitro Cytotoxic Assay

HepG2 (human liver cancer) and HT29 (human colon and colorectal adenocarcinoma) cell lines were used to assess the toxicity of samples. In the performed assay, cytotoxicity was expressed as a concentration-dependent reduction in the uptake of the vital dye Neutral Red (NR) when measured 24 h after treatment. NR is a weak cationic dye that readily penetrates cell membranes by non-diffusion and accumulates intracellularly in lysosomes. Alterations of the sensitive lysosomal membrane lead to lysosomal fragility and other changes that gradually become irreversible. This results in a decreased uptake and binding of NR in non-viable cells. HT29 (ATCC^®^ HTB-38™) and HepG2 (ATCC^®^ HB-8065™), low passage number (<50), were cultivated into DΜΕΜ (Dulbecco’s Minimum Essential Medium, PAN BIOTECH. lot 1874561) supplemented with penicillin 100 IU/mL and streptomycin 100 μg/mL (PAN BIOTECH, Lot 945514), and 10% of inactivated calf serum (PAN BIOTECH, Lot P56314), pH 7.2, freshly prepared, stored no longer than 1 week. Cells were seeded into 96-well tissue culture plates (0.1 mL per well) at a concentration of 1.10^5^ cells/mL and incubated at 37 °C (5% CO_2_) until semi-confluent. The test material was diluted into sterile DMSO (stock solutions 0.1, 1 and 10 mg/mL) at final concentrations ranging from 0.1 to 250 µg/mL. The culture medium was decanted and replaced by 100 µL of fresh medium containing the various concentrations of the test material; then, cells were incubated for 24 h at 37 °C (5% CO_2_). At the end of the incubation period, cells were placed into Neutral Red medium (50 μg/mL NR in complete medium) and incubated for 3 h at 37 °C, 5% CO_2_. Then, the medium was removed, and cells were washed three times with 0.2 mL of HBSS to remove excessive dye. The Neutral Red medium was removed and the distaining solution (50% ethanol, 1% acetic acid, 49% distilled water; 50 µL per well) was added into the wells. Then, the plates were shaken for 15–20 min at room temperature in the dark. The test samples and controls were run in triplicates in three independent experiments. A fluorescence–luminescence reader Infinite M200 Pro (TECAN) was used to measure the degree of membrane damage (i.e., the increase in released NR). For each well, the Optical Density (OD) was read at 540 nm. The results obtained for test material wells were compared to those of untreated control wells (HBSS, 100% viability) and converted to percentage values. The concentrations of the test material causing a 50% release of the preloaded NR (IC_50_) compared to the control culture were calculated using software Phototox Version 2.0. The mean OD value of blank wells (only NR desorbed solution) was subtracted from the mean OD value of three test/untreated wells.

### 2.6. Feature-Based Molecular Networking

The leaf crude extract of *C. tacamahaca* as well as the isolated metabolites were profiled by UHPLC-QqTOF-MS/MS in a mass range from *m*/*z* 50 to 1200 using positive (+) mode for the ESI source. The following parameters were used: end plate offset at 500 V; nebulizer gas pressure at 3.5 bar; dry gas flow at 12 L/min; drying temperature at 200 °C; acquisition rate at 4.0 Hz. The capillary voltage was set at 4500 V, with a fragmentation energy of 20–40 eV. The UHPLC conditions were as follows: sample concentrations: 5 mg/mL (crude extract), 0.2 mg/mL (isolated compounds) in 100% MeOH, injection volume: 2 µL, column temperature: 40 °C, elution gradient of H_2_O-CH_3_CN with 0.1% HCO_2_H (98:02 over 2 min, 98:02 to 0:100 over 12 min, 0:100 over 3 min) at a flow rate of 0.5 mL/min. Raw data obtained from the crude extract analysis were converted into open format .mzXML using software Bruker Compass DataAnalysis Version 4.2 and processed using software MZmine Version 2.53 [19,20,21]. Then, a feature-based molecular network (FBMN) was created on the GNPS platform [22], and it is available via the following link https://gnps.ucsd.edu/ProteoSAFe/status.jsp?task=f0c193d2141d463ba34af46df7bfe57c (accessed on 29 March 2022). The Mzmine MS/MS data processing comprises .mzXML file import, MS peak detection, ADAP chromatogram builder, chromatogram deconvolution, isotopic peaks grouper, alignment, filtering, fragment search, adduct search and spectra normalization. Setting parameters were as follows: positive ionization mode, centroid detection, MS1 peak detection limit: 1^E^3, MS2 peak detection limit: 1^E^2, *m*/*z* tolerance: 10 ppm, peak/top edge ratio: 2, peak duration range: 0.03–1 min*, m*/*z* range for MS2 pairing: 0.02 Da, RT range for MS2 pairing: 0.1 min, representative isotope: most intense, alignment weight for *m*/*z*: 75, weight for RT: 25, filtering RT tolerance: 0.1 min, filtering *m*/*z* tolerance: 0.001 *m*/*z*, adduct search [M+Na]^+^, [M+NH_4_]^+^, spectra normalization type: average intensity. Processed files including an mgf and a csv file were uploaded to the GNPS platform. An FBMN was then developed using the Advanced Analysis Tools—Feature Networking workflow [23]. Advanced Network Options parameters were as follows: min pair cos: 0.7, minimum matched fragment ions: 6, network topK: 10, maximum connected component size: 100, mass tolerance for precursor and fragment ions: 0.02 Da. The output was imported into Cytoscape Version 3.8.2 in order to visualize the network. Node annotations were performed manually for isolated compounds and with GNPS spectral databases (score threshold: 0.7) and In Silico MS/MS DataBase ISDB (score threshold: 0.2) [24].

## 3. Results and Discussion

### 3.1. Isolation of Compounds **1**–**14**

C. tacamahaca leaf EtOAc extract was subjected to a solid reverse-phase extraction and yielded three fractions (F1–F3). Fractions F2 and F3 were further purified by preparative, semi-preparative and analytical reverse-phase HPLC, resulting in the isolation of one new chromanone (**1**) and 13 known compounds (**2**–**14**) (Figure 1). The latter were identified by comparison with previously reported spectroscopic data as amentoflavone (**2**) [25], scriblitifolic acid (**4**) [26], pancixanthone B (**5**) [27], calophyllic acid (**6**) [28] isocalophyllic acid (**7**) [28], inophyllum E (**8**) [28,29], 27-hydroxyacetate-canophyllic acid (**9**) [30], pyranojacareubin (**10**) [31], canophyllalic acid (**11**) [32], canophyllol (**12**) [32], canophyllic acid (**13**) [32] and canophyllal (**14**) [33]. Spectroscopic data of the known metabolite **3**, identified as 6-(4-hydroxy-3-methylbutyl)-1,5-dihydroxyxanthone [34], have not been published so far and are provided here (Section 2 and Appendix A). The structure of the new compound **1** was established based on 1D and 2D NMR, IR and UV spectroscopic and HRESIMS spectrometric data.

### 3.2. Structure Elucidation of Isocaloteysmannic Acid (**1**)

Isocaloteysmannic acid (**1**), ${\left[\alpha \right]}_{D}^{25}$ −31.7 (c 0.1, MeOH), was isolated as a yellow–green powder. The molecular formula C_25_H_26_O_6_ was established from HRESIMS data showing a molecular ion peak at *m*/*z* 423.1791 [M+H]^+^ (calculated for C_25_H_27_O_6_^+^, 423.1802), suggesting the occurrence of 13 degrees of insaturation. The UV spectrum exhibited absorption maxima at 200, 264–274, 299–312 and 368 nm, characteristic of a pyranochromanone moiety [35]. The IR spectrum exhibited characteristic bands of sp^3^ type CH (2926 cm^−1^), sp^2^ type CH (3087 cm^−1^), carboxylic acid function (1709 cm^−1^), aromatic rings (1627 cm^−1^) and ether function (1000 and 1300 cm^−1^). The ^1^H and ^13^C NMR data of (**1**) (Table 1 and Appendix A) are similar to those of caloteysmannic acid [35]. The ^1^H and ^13^C NMR spectra showed aromatic signals at δ_H/C_ 7.33 (H-2′, H-6′, doublet)/128.8 (C-2′, C-6′), δ_H/C_ 7.20 (H-3′, H-5′, borad triplet)/128.8 (C-3′, C-5′) and δ_H/C_ 7.10 (H-4′, triplet)/126.7 (C-4′), consistent with the phenyl group of the chromanone. The COSY spectrum (Appendix A) showed correlations consistent with the spin system H-2′−H-3′−H-4′−H-5′−H-6′. Signals observed at δ_H/C_ 5.48 (H-7, doublet)/127.3 (C-7) and δ_H/C_ 6.49 (H-8, doublet)/116.7 (C-8) correspond to the spin-pair of two sp^2^ methine protons. Characteristic signals of protons H-10 and H-11 are observed at δ_H/C_ 4.18 (H-10, doublet of quadruplet)/80.3 (C-10) and δ_H/C_ 2.61 (H-11, doublet of quadruplet)/46.9 (C-11). The two signals observed at δ_H/C_ 3.07 (H-3a, doublet of doublet)/38.2 (C-3) and 3.27 (H-3b, doublet of doublet)/38.2 (C-3) correspond to diastereotopic protons. The ^1^H and ^13^C NMR spectra show a deshielded signal at δ_H/C_ 5.07 (H-4, broad triplet)/36.3 (C-4), corresponding to the alkane proton in beta position of the acid carboxylic function. These positions were confirmed with the COSY spectrum (Appendix A) showing a correlation between H-3 (δ_H_ 3.07, 3.27) and H-4 (δ_H_ 5.07). Four signals corresponding to methyl groups are observed at δ_H/C_ 1.01 (H-13, singlet)/27.5 (C-13), δ_H/C_ 1.41 (H-14, singlet)/28.5 (C-14), δ_H/C_ 1.49 (H-15, doublet)/19.8 (C-15) and δ_H/C_ 1.19 (H-16, doublet)/10.3 (C-16). The COSY spectrum (Appendix A) shows correlations between H-15 (δ_H_ 1.49) and H-10 (δ_H_ 4.18), and between H-16 (δ_H_ 1.19) and H-11 (δ_H_ 2.61). Finally, the characteristic signals of the acid carboxylic and the ketone functions are observed on the ^13^C NMR spectrum at δ_C_ 177.2 (C-2) and δ_C_ 200.7 (C-12), respectively. The linkage and the substitution pattern of (**1**) is determined from HMBC correlations (Figure 2 and Appendix A). The HMBC correlations of H-4 (δ_H_ 5.07) to C-2′ (δ_C_ 128.8) and C-6′ (δ_C_ 128.8) and those of H-3 (δ_H_ 3.07, 3.27) to C-1′ (δ_C_ 145.2) indicate the substitution of C-4 (δ_C_ 36.3) by the phenyl group. The carboxylic acid function position in C-2 (δ_C_ 177.2) is confirmed by the ^2^J_HC_ correlation of H-3 (δ_H_ 3.07, 3.27) to C-2 (δ_C_ 177.2). The HMBC correlations of methyl protons H-13 (δ_H_ 1.01) and H-14 (δ_H_ 1.41) to C-6 (δ_C_ 79.2) indicate these two methyl groups are borne by the same carbon C-6 (δ_C_ 79.2). The HMBC correlations of H-7 (δ_H_ 5.48) to C-14 (δ_C_ 28.5) and C-8a (δ_C_ 102.9), and of H-8 (δ_H_ 6.49) to C-4b (δ_C_ 160.9), C-6 (δ_C_ 79.2) and C-8b (δ_C_ 156.8) confirmed the A and C rings linkage. The HMBC correlations of H-4 (δ_H_ 5.07) to C-4b (δ_C_ 160.9) and C-12b (δ_C_ 162.2), and of H-3 (δ_H_ 3.07, 3.27) to C-4a (δ_C_ 113.0) indicate the substitution of C-4a (δ_C_ 113.0) by the phenyl-bearing saturated chain. Finally, the HMBC correlations of H-10 (δ_H_ 4.18) and H-11(δ_H_ 2.61) to C-16 (δ_C_ 10.3) and C-15 (δ_C_ 19.8), respectively, of H-10 (δ_H_ 4.18) to C-12 (δ_C_ 200.7) and C-8b (δ_C_ 156.8), and those of H-16 (δ_H_ 1.19) to C-12 (δ_C_ 200.6) confirm the D ring configuration. Based on NMR data, the ^3^J_H-10/11_ coupling constant (11.3 Hz) between the vicinal protons H-10 and H-11 indicate a dihedral angle consistent with an axial–axial coupling constant [36]. In a previous work, Patil et al. showed that the only possible configuration for these trans-diaxial H-10 and H-11 vicinal protons is a configuration of C-10 and C-11 carbons 10R, 11R [28]. Consequently, two potential diastereoisomers were conceivable for compound **1**: (4*R*,10*R*,11*R*) or (4*S*,10*R*,11*R*) (Figure 3).

### 3.3. Absolute Configuration of Isocaloteysmannic Acid (**1**)

First, to confirm the configuration of C-10 and C-11 carbons and to determine the configuration of C-4 carbon, experimental chemical shifts (^1^H and ^13^C) of compound (**1**) were compared with calculated chemical shifts of four isomers (4*S*,10*S*,11*S*), (4*S*,10*R*,11*R*), (4*S*,10*R*,11*S*), (4*S*,10*S*,11*R*). For these four isomers, the equilibrium population of each conformer was calculated from its relative free energy using Boltzmann statistics, 36 conformers for (4*S*,10*S*,11*S*), 37 conformers for (4*S*,10*R*,11*R*), 39 conformers for (4*S*,10*R*,11*S*), and 48 conformers for (4*S*,10*S*,11*R*) (Appendix A). NMR chemical shifts have been calculated with the GIAO method at the PCM/mPW1PW91/6-31+G(d,p) level allowing to use the DP4+ probability [37]. Experimental chemical shifts have been compared to theorical chemical shifts of each isomer individually by linear regressions of *δ*^1^H_theorical_ = f(*δ*^1^H_experimental_) and *δ*^13^C_theorical_ = f(*δ*^13^C_experimental_) and all together with the DP4+ probability (Appendix A). Assignment by ^1^H-DP4+ and ^13^C-DP4+ did not converge to the same isomer, and when including all the data, probabilities were shared between two isomers (4*S*,10*S*,11*S*) (41.07%) and (4*S*,10*R*,11*R*) (58.93%) (Figure 4). Therefore, the results of these comparisons did not allow unambiguous determination of the absolute configuration of compound **1** but did confirm the trans-configuration of C-10 and C-11.

The absolute configuration of (**1**) was established by ECD by comparing the measured spectra with those calculated using DFT and TD-DFT for diastereomers (4*S*,10*R*,11*R*) and (4*R*,10*R*,11*R*) according to the previous NMR analysis (Figure 3).

The UV and ECD spectra of (4*S*,10*R*,11*R*) and (4*R*,10*R*,11*R*) were built, respectively, from the individual spectra of the A_1–6_ and B_1–6_ conformations weighed by their Boltzmann population (Appendix B). The comparison of the calculated UV spectra for the two diastereomers showed a good agreement with the measured spectrum, without allowing to establish the absolute configuration of the C-4 atom. Furthermore, the calculated ECD spectra showed a clear sign difference around 215 nm: positive bands for (4*S*,10*R*,11*R*) and negative bands for (4*R*,10*R*,11*R*) (Figure 5A–D).

Comparison with the corresponding measured spectrum showed excellent agreement with that calculated for the (4*S*,10*R*,11*R*) configuration (Figure 5A–D). In particular, the band around 215 nm is positive as in the measured spectrum. This ECD analysis therefore confirmed the *R*-configuration of the C-10 and C-11 atoms, but also unambiguously established that the C-4 atom is of absolute configuration *S*. Consequently, compound **1** has the absolute configuration (4*S*,10*R*,11*R*).

Compound **1** is a trans-epimer of caloteysmannic acid, a chromanone with (4*S*) configuration and cis-configuration of vicinal protons H-10, H-11 (10*S*,11*R*) previously isolated from *Calophyllum teysmannii* [35]. Therefore, (**1**) was named isocaloteysmannic acid.

### 3.4. Cytotoxic Activity of the Isolated Compounds

Ten isolated compounds were evaluated for their cytotoxic properties against the two cancer cell lines HepG2 and HT29. Due to their paucity, compounds **3**–**5** and **14** were not evaluated. Compounds **7**, **8**, **10**, **11**, **12** and **13** showed a potent activity against one or both cell lines, with IC_50_ values ranging from 2.44 to 15.38 µg/mL (Table 2). The new compound **1**, as well as compound **6**, exhibited a moderate activity against both cell lines with IC_50_ values ranging from 15.98 to 25.68 µg/mL.

The triterpenes **11**–**13** showed a potent activity, whereas triterpene **9** exhibited only a weak activity, suggesting that the presence of the acetoxy group in **9** could decrease its cytotoxic potential.

These results also suggest that the cis-configuration of the methyl groups in C-10 and C-11 of compounds **7** and **8** leads to a higher cytotoxic activity than the trans-configuration (compounds **1** and **6**).

### 3.5. Feature-Based Molecular Networking Analysis of the Crude Extract

A feature-based molecular networking (FBMN) [23] approach was performed in order to provide more information about the chemodiversity of the species and to detect additional cytotoxic metabolites by highlighting close analogues of the bioactive isolated compounds. For this purpose, leaf EtOAc extract was subjected to an UHPLC-HRESIMS/MS analysis and a molecular network (MN) was generated with the FBMN tool on the GNPS platform.

#### 3.5.1. Chemodiversity of the Species

A molecular network (MN) comprising 520 features and 55 clusters (two features at least) was obtained (Figure 6). Squared orange nodes correspond to the isolated compounds **1**, **3**, **4**, **5**, **6**, **8** and **10**. Green nodes correspond to spectral matches on GNPS or ISDB databases. The edge thickness correlates with the cosine score (CS) value (0.7–1) between two nodes.

Relatively few consistent spectral matches on GNPS or ISDB databases were obtained. Based on these matches, the largest cluster C1 (43 nodes) could correspond to xanthones. Two nodes correspond to the isolated xanthones **5** and **10**, and three nodes were putatively identified as xanthones previously reported in the genus *Calophyllum*: 6-deoxyisojacareubin, mammea B/BA and caloxanthone. Seven nodes could correspond to xanthones reported in close botanical families of Calophyllaceae: elliptoxanthone B (Hypericaceae), garcinexanthone C (Clusiaceae), celebixanthone (Hypericaceae), nigrolineaxanthone K (Clusiaceae), garcinone A (Clusiaceae), hypericumxanthone B (Hypericaceae) and garcimangosone C (Clusiaceae).

Cluster C8 is another cluster of xanthones, containing the isolated metabolites **3** and **4**, as well as one node putatively identified as caloxanthone H. The latter was previously reported in the genus *Calophyllum*.

The new compound **1** is located in cluster C5. In the latter, one node corresponds to a close analogue of **1** (*m*/*z* 423.1785, CS > 0.9). Based on ISDB matches, this close analogue was putatively identified as isochapelieric acid, a compound isolated from the species *Calophyllum calaba* [38].

These observations are consistent with the data in the literature, indicating that xanthones and chromanones are largely represented in the genus *Calophyllum*.

#### 3.5.2. Detection of Additional Bioactive Metabolites

Two analogues of the cytotoxic isolated compound pyranojacareubin (**10**) have been detected in cluster C1 (Figure 7) at *m*/*z* 395.1475 and *m*/*z* 327.0854. Based on structure–activity relationship, these analogues could correspond to cytotoxic metabolites. They were putatively identified as muxiangrine I and elliptoxanthone B, according to ISDB matches. To the best of our knowledge, no cytotoxic properties have been reported in the literature for these compounds. As these identifications are highly hypothetical, it would be necessary to target the isolation of these two compounds, to identify them and assess their biological properties in a future work.

## 4. Conclusions

Fourteen metabolites (**1**–**14**) were isolated from the EtOAc leaf extract of *C. tacamahaca*. To the best of our knowledge, compound **1** was reported for the first time. Six compounds (**7**, **8**, **10**, **11**, **12** and **13**) showed a potent cytotoxicity against HepG2 and/or HT29 cell lines. The FBMN approach allowed the detection of a large amount of xanthones in the extract, including two close analogues of the cytotoxic compound **10**. Xanthones are well known for their cytotoxic properties [2], so the results of this study suggest that *C. tacamahaca* leaves are a significant source of cytotoxic metabolites. These compounds could be interesting candidates for future therapeutic applications. Nevertheless, further studies are needed to evaluate their in vivo anticancer activity, as well as their mechanism of action, and thus confirm their therapeutic potential.

## Figures and Tables

**Figure 1 metabolites-13-00582-f001:**
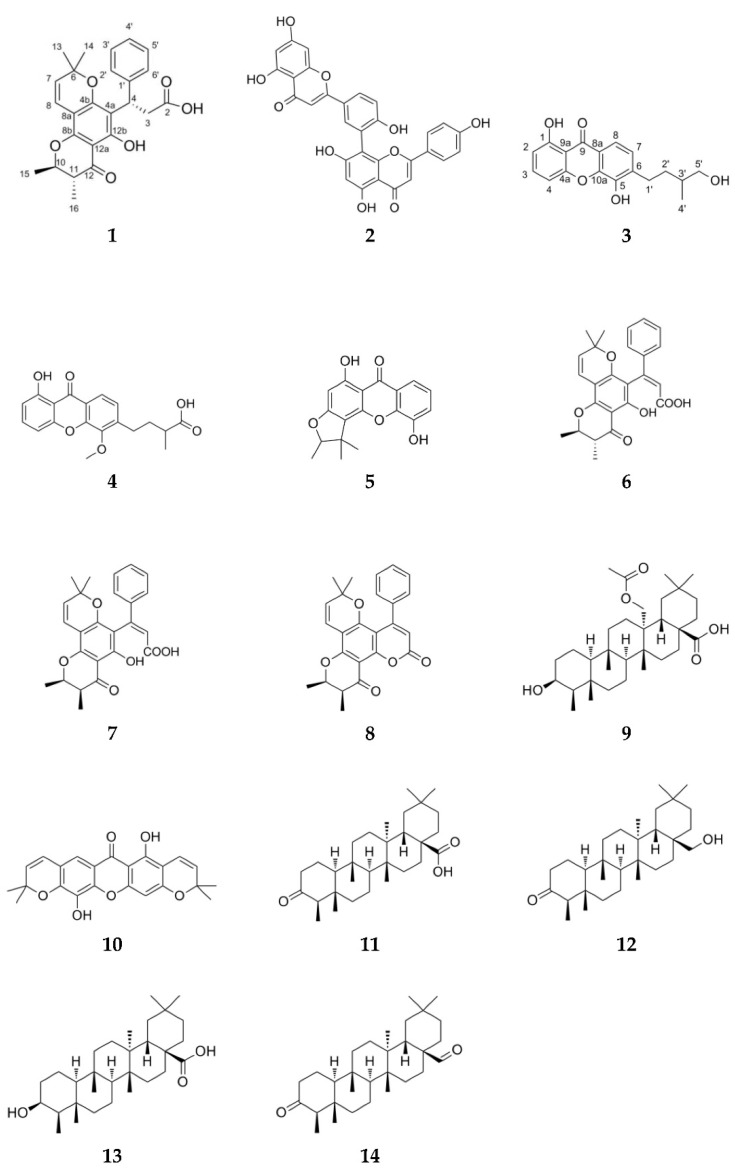
Structures of compounds **1**–**14** isolated from *C. tacamahaca*.

**Figure 2 metabolites-13-00582-f002:**
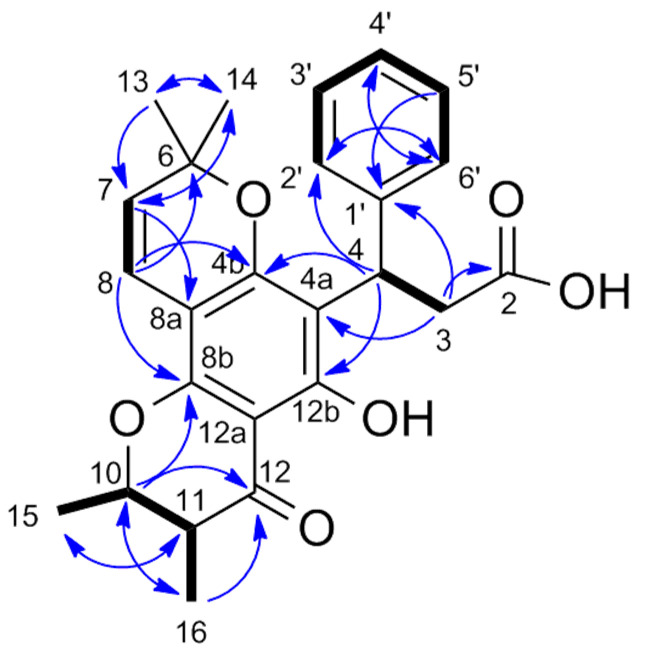
Key ^1^H-^1^H COSY (bold) and ^1^H-^13^C HMBC (blue arrows) correlations of (**1**).

**Figure 3 metabolites-13-00582-f003:**
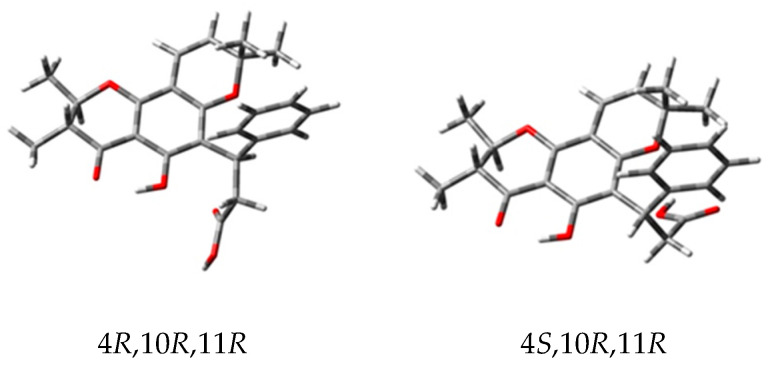
The two potential enantiomers for (**1**).

**Figure 4 metabolites-13-00582-f004:**
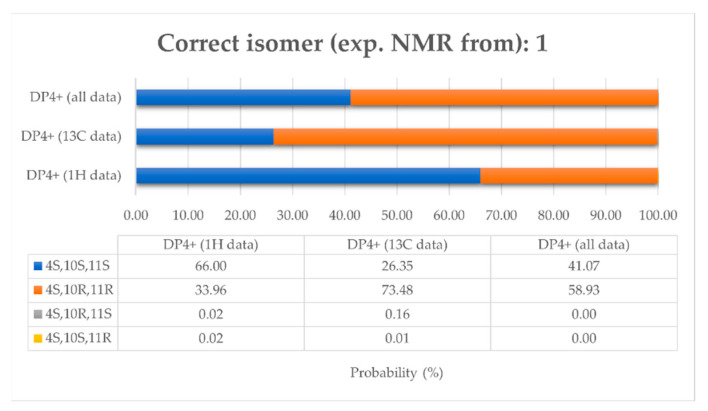
Graph of ^1^H-DP4+, ^13^C-DP4+, and DP4+ (PCM/mPW1PW91/6-31+Gdp) probabilities obtained by correlating the experimental NMR of compound **1** with the calculated data of the four isomers (4*S*,10*S*,11*S*), (4*S*,10*R*,11*R*), (4*S*,10*R*,11*S*), (4*S*,10*S*,11*R*).

**Figure 5 metabolites-13-00582-f005:**
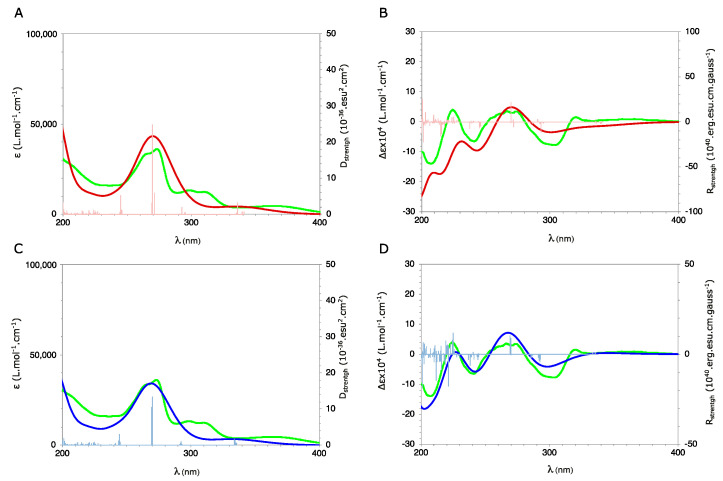
UV (**left-A**,**C**) and ECD (**right-B**,**D**) spectra measured in CD_3_OD for (**1**) (green) and calculated using SMD(CH_3_OH)/CAM-B3LYP/6-31++G(d,p)//GD3BJ-B3LYP/6-311G(d,p) level for (4*R*,10*R*,11*R*) (red) and (4*S*,10*R*,11*R*) (blue).

**Figure 6 metabolites-13-00582-f006:**
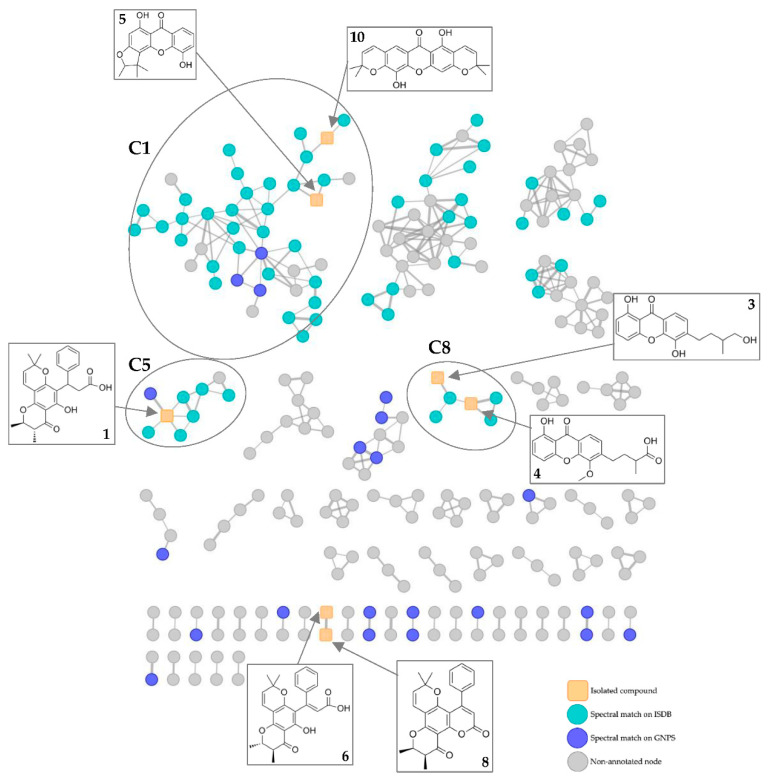
Molecular networking (MN) of the isolated compounds (orange squared nodes) and MN annotation based on GNPS and ISDB spectral matches (green and blue nodes).

**Figure 7 metabolites-13-00582-f007:**
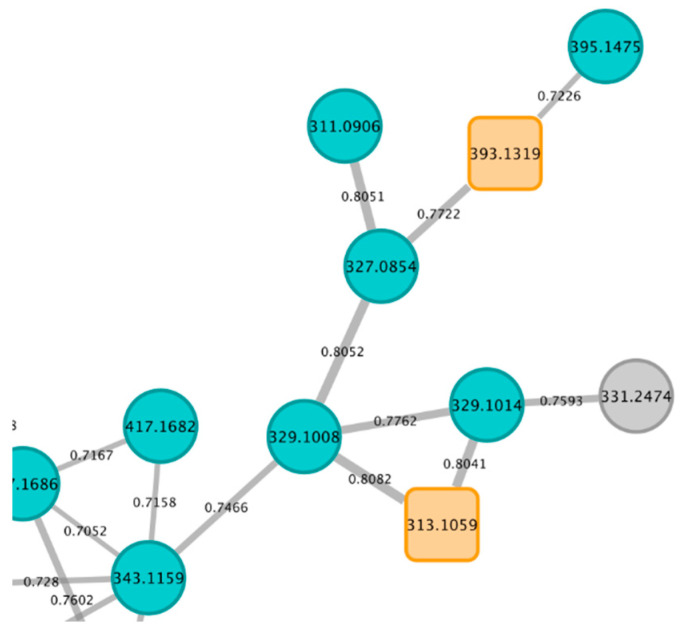
Part of cluster C1 containing analogues of compound **10** (orange squared node at *m*/*z* 393.1319). Ion parent mass is indicated in each node and cosine score value is indicated on each edge.

**Table 1 metabolites-13-00582-t001:** ^1^H and ^13^C NMR data of isocaloteysmannic acid (**1**) in CD_3_OD (600 MHz for ^1^H and 150 MHz for ^13^C).

Position	δ_H_ m (*J* in Hz)	δ_C_
2	-	177.2
3	3.07, dd (15.2, 7.2) 3.27, dd (15.2, 8.2)	38.2
4	5.0, 7 brt ^a^ (7.7)	36.3
4a	-	113.0
4b	-	160.9
6	-	79.2
7	5.48, d (10.0)	127.3
8	6.49, d (10.0)	116.7
8a	-	102.9
8b	-	156.8
10	4.18, dq (11.3, 6.2)	80.3
11	2.61, dq (11.3, 6.9)	46.9
12	-	200.7
12a	-	102.6
12b	-	162.2
13	1.01, s	27.5
14	1.41, s	28.5
15	1.49, d (6.2)	19.8
16	1.19, d (6.9)	10.3
1′	-	145.2
2′, 6′	7.33, d (7.6)	128.8
3′, 5′	7.20, brt (7.5)	128.8
4′	7.10, brt (7.3)	126.7

^a^ br: broad.

**Table 2 metabolites-13-00582-t002:** Cytotoxic activity of the isolated compounds.

Compound	IC_50_ (µg/mL) ^a^	
	HepG2	HT29
**1**	19.65 ± 2.34	25.68 ± 2.08
**2**	39.03 ± 3.23	41.97 ± 2.54
**6**	15.98 ± 3.65	18.97 ± 2.94
**7**	2.44 ± 0.67	4.24 ± 0.67
**8**	7.03 ± 1.56	5.94 ± 0.07
**9**	45.09 ± 2.09	56.98 ± 3.76
**10**	9.54 ± 1.22	10.46 ± 2.08
**11**	3.34 ± 0.94	5.97 ± 0.99
**12**	15.38 ± 2.07	10.26 ± 1.34
**13**	6.65 ± 1.54	4.06 ± 0.29

^a^ IC_50_ are the means ± standard deviations calculated from three independent assays.

## Data Availability

NMR raw data (^1^H, ^13^C, gCOSY, gHSQC, gHMBC) of compounds **1** and **3** are made freely available at https://doi.org/10.5281/zenodo.7728239. Raw MS/MS data (open format .mzXML) have been deposited on MassIVE (https://massive.ucsd.edu): MSV000089771.

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
