# Peer review of "Cytotoxic Metabolites from *Calophyllum tacamahaca* Willd.: Isolation and Detection through Feature-Based Molecular Networking"

_metabolites, 2023, doi:10.3390/metabo13050582_

Round 1

Reviewer 1 Report

The manuscript reports on isolation and structure elucidation of a new compound isocaloteydmannic acid from Calophyllum tacamachaca, along with 13 known molecules. Cytotoxicity test was done on two cell lines and some known compounds showed good activity. In addition, molecular networking analysis revealed potential bioactive analogues for further investigation. Overall the study design and methodologies employed are appropriate, and structure elucidation and conformation determination appeared to be solid. The manuscript was well organized with good quality including supplemental information.

Question/comment:

Isolation of both cis and trans C10/C11 isomers was found in this work and some cited papers. Please consider to add a paragraph commenting/elaborating whether they are all real natural products (biosynthetically produced), or one of them might be an artifact.  

Author Response

Isolation of both cis and trans C10/C11 isomers was found in this work and some cited papers. Please consider to add a paragraph commenting/elaborating whether they are all real natural products (biosynthetically produced), or one of them might be an artifact.  

Both cis and trans C10/C11 isomers are natural products. The cis isomer has been isolated from the stem bark of the species Calophyllum tesymannii by Lim et al., 2015. The trans isomer is reported here for the first time, and has been isolated from Calophyllum tacamahaca.

In p. 13, we mentioned that « Compound 1 is a trans-epimer of caloteysmannic acid, a chromanone with (4S) configuration and cis configuration of vicinal protons H-10, H-11 (10S,11R) previously isolated from Calophyllum teysmannii [35]. »

Reviewer 2 Report

In this manuscript, the authors extracted some new compounds from Calophyllum tacamahaca Willd. A new compound was identified, and its cytotoxicity was checked on two cell lines. The manuscript is well-written and can be published after some minor revisions. 

1. In the introduction, it was mentioned that the plant is registered in French pharmacopeia. For which application and what part of the plant?

2. Methanol and ethanol are commonly used extractors. Why ethyl acetate has been chosen?

3. The author mentioned that the compounds from chosen plants have never been isolated and characterized. Why compounds 2-14 are already known then?

4. Please assign all the peaks in NMR to the structure to get a clear picture. Also, include multiple analyses with reports in the method section for each isolate characterized by NMR.

5. Please include mass spec for compound 3.

6. Please expand the conclusion by including an outlook on the possible applications of extracted compounds. The majority of compounds are cytotoxic to cancer cells, but that does not really mean that they have anticancer activity. A mechanism of cell killing is warranted to conclude the anticancer activity. 

Author Response

1. In the introduction, it was mentioned that the plant is registered in French pharmacopeia. For which application and what part of the plant?

The leaf is registered in French Pharmacopoeia, for eye diseases, against fever, headaches and as veinotonic. We added this information in the introduction.

2. Methanol and ethanol are commonly used extractors. Why ethyl acetate has been chosen?

We chose to use ethyl acetate because this solvent allows to extract a wide diversity of compounds, from mildly polar to apolar compounds.

3. The author mentioned that the compounds from chosen plants have never been isolated and characterized. Why compounds 2-14 are already known then?

If this question refers to the following statement (p. 2): « Nevertheless, the chemical composition of the species has never been studied and so bioactive compounds of the species have never been isolated nor identified so far”, we meant this was the first time compounds were isolated and identified from the species Calophyllum tacamahaca, but that did not mean these compounds were necessarily knew ones. Some of them (2-14) had been previously described in other species.

4. Please assign all the peaks in NMR to the structure to get a clear picture. Also, include multiple analyses with reports in the method section for each isolate characterized by NMR.

We are not sure to correctly understand your request. We added NMR, UV and HRMS data of all described compounds in the Method section.

5. Please include mass spec for compound 3.

Mass spectrum of compound 3 is in Supporting info file (Figure S11).

6. Please expand the conclusion by including an outlook on the possible applications of extracted compounds. The majority of compounds are cytotoxic to cancer cells, but that does not really mean that they have anticancer activity. A mechanism of cell killing is warranted to conclude the anticancer activity. 

This sentence was added to the conclusion : « These compounds could be interesting candidates for future therapeutic applications. Nevertheless, further studies are needed to evaluate their in vivo anticancer activitiy, as well as their mechanism of action, and thus confirm their therapeutic potential. »

Reviewer 3 Report

The manuscript "Cytotoxic metabolites from Calophyllum tacamahaca Willd.: isolation and detection through Feature-Based Molecular Networking", shows important data for C. tacamahaca. However, the authors should clarify the following points:

- Although the authors indicate that it is a plant species, and there are reports of biological activities (Antioxidants: doi= 10.3390/antiox10020199) of aqueous or hydroalcoholic extracts of C. tacamahaca. Because it was decided to perform extractions with ethyl acetate.

- What was the criteria for the separation and isolation of the compounds from the ethyl acetate extract? Was it biodirected?

- The authors determine the cytotoxicity of the compounds on tumor cell lines. However, this would be a possible antitumor activity. The correct thing is to determine the cytotoxicity on non-tumor cell lines. So, what was the criteria for choosing these two cell lines?

- Positive controls (drugs) must be added in biological assays to be able to compare the results obtained.

Author Response

1. Although the authors indicate that it is a plant species, and there are reports of biological activities (Antioxidants: doi= 10.3390/antiox10020199) of aqueous or hydroalcoholic extracts of C. tacamahaca. Because it was decided to perform extractions with ethyl acetate.

We chose to use ethyl acetate because this solvent allows to extract a wide diversity of compounds, from mildly polar to apolar compounds. More, some interesting activities (antiparasite, antimicrobial) were reported for ethyl acetate extracts of the species.

2. What was the criteria for the separation and isolation of the compounds from the ethyl acetate extract? Was it biodirected?

Yes, the purification of the compounds was bio-guided. We added this information in the revised version (p. 2).

3. The authors determine the cytotoxicity of the compounds on tumor cell lines. However, this would be a possible antitumor activity. The correct thing is to determine the cytotoxicity on non-tumor cell lines. So, what was the criteria for choosing these two cell lines?

One of the aims of the study was to identify the bioactive metabolites of the species, i.e. compounds with a potential anticancer activity. Thus, we chose two cancer cell lines for the in vitro bioactivity tests: liver cancer cell line (HepG2) and colorectal cancer cell line (HT29).

4. Positive controls (drugs) must be added in biological assays to be able to compare the results obtained.

There is no attested standard drug for the cytotoxic activity. Interpration of the results was performed following the recommendations of Bero et al., 2010 and Quetin-Leclercq 2011:

- IC50 value ≤ 15 µg/mL: good activity

- 15 µg/mL  < IC50 value < 50 µg/mL: moderate activity

- 50 µg/mL < IC50 value: no activity